# A small protein encoded by a putative lncRNA regulates apoptosis and tumorigenicity in human colorectal cancer cells

Xiao Ling Li[1], Lőrinc Pongor[2], Wei Tang[3], Sudipto Das[4], Bruna R Muys[1], Matthew F Jones[1], Sarah B Lazar[1], Emily A Dangelmaier[1], Corrine CR Hartford[1], Ioannis Grammatikakis[1], Qinyu Hao[5], Qinyu Sun[5], Aaron Schetter[6], Jennifer L Martindale[7], BinWu Tang[8], Lisa M Jenkins[9], Ana I Robles[6], Robert L Walker[10], Stefan Ambs[3], Raj Chari[11], Svetlana A Shabalina[12], Myriam Gorospe[7], S Perwez Hussain[13], Curtis C Harris[6], Paul S Meltzer[10], Kannanganattu V Prasanth[5], Mirit I Aladjem[2], Thorkell Andresson[4], Ashish Lal[1]*

[1]Regulatory RNAs and Cancer Section, Genetics Branch, Center for Cancer Research (CCR), National Cancer Institute (NCI), National Institutes of Health (NIH), Bethesda, United States; [2]Developmental Therapeutics Branch, CCR, NCI, NIH, Bethesda, United States; [3]Molecular Epidemiology Section, Laboratory of Human Carcinogenesis, CCR, NCI, NIH, Bethesda, United States; [4]Protein Characterization Laboratory, Cancer Research Technology Program, Frederick National Laboratory for Cancer Research, Leidos Biomedical Research, Inc, Frederick, United States; [5]Department of Cell and Developmental Biology, Cancer Center at Illinois University of Illinois at Urbana-Champaign, Urbana, United States; [6]Molecular Genetics and Carcinogenesis Section, Laboratory of Human Carcinogenesis, CCR, NCI, NIH, Bethesda, United States; [7]Laboratory of Genetics and Genomics, National Institute on Aging Intramural Research Program, NIH, Baltimore, United States; [8]Laboratory of Cancer Biology and Genetics, CCR, NCI, NIH, Bethesda, United States; [9]Laboratory of Cell Biology, CCR, NCI, NIH, Bethesda, United States; [10]Molecular Genetics Section, Genetics Branch, CCR, NCI, NIH, Bethesda, United States; [11]Genome Modification Core, Frederick National Lab for Cancer Research, National Cancer Institute, Frederick, United States; [12]National Center for Biotechnology Information, National Library of Medicine, NIH, Bethesda, United States; [13]Pancreatic Cancer Unit, Laboratory of Human Carcinogenesis, CCR, NCI, NIH, Bethesda, United States

*For correspondence:
ashish.lal@nih.gov

**Abstract** Long noncoding RNAs (lncRNAs) are often associated with polysomes, indicating coding potential. However, only a handful of endogenous proteins encoded by putative lncRNAs have been identified and assigned a function. Here, we report the discovery of a putative gastrointestinal-tract-specific lncRNA (*LINC00675*) that is regulated by the pioneer transcription factor FOXA1 and encodes a conserved small protein of 79 amino acids which we termed FORCP (*FOXA1-Regulated Conserved Small Protein*). *FORCP* transcript is undetectable in most cell types but is abundant in well-differentiated colorectal cancer (CRC) cells where it functions to inhibit proliferation, clonogenicity, and tumorigenesis. The epitope-tagged and endogenous FORCP protein predominantly localizes to the endoplasmic reticulum (ER). In response to ER stress, *FORCP* depletion results in decreased apoptosis. Our findings on the initial characterization of *FORCP*

demonstrate that FORCP is a novel, conserved small protein encoded by a mis-annotated lncRNA that regulates apoptosis and tumorigenicity in well-differentiated CRC cells.

## Introduction

The human genome harbors thousands of long noncoding RNA (lncRNA) genes transcribed into a heterogeneous group of transcripts > 200 nucleotides (nt) long (*Harrow et al., 2012*; *Zhao et al., 2016*). Although the vast majority of lncRNAs have not been functionally characterized, some have established functions in development and disease (*Fatica and Bozzoni, 2014*; *Li and Chang, 2014*) and control diverse cellular processes including dosage compensation, differentiation, proliferation, apoptosis, metastasis, DNA repair, and genome stability maintenance (*Fatica and Bozzoni, 2014*; *Lee et al., 2016*; *Munschauer et al., 2018*; *Sharma et al., 2015*; *Arun et al., 2016*; *Sahakyan et al., 2018*). lncRNA expression is frequently altered in cancer (*Prensner and Chinnaiyan, 2011*; *Huarte and Rinn, 2010*; *Huarte, 2015*). We and others have shown that some lncRNAs play important roles in mediating the effects of tumor-suppressive transcription factors such as p53 (*Chaudhary et al., 2017*; *Li et al., 2017*; *Adriaens et al., 2016*; *Hünten et al., 2015*; *Léveillé et al., 2015*; *Grossi et al., 2016*; *Marín-Béjar et al., 2017*; *Huarte et al., 2010*; *Chaudhary and Lal, 2017*). Investigating the molecular and cellular functions of lncRNAs could uncover their roles in cancer and lay the foundation for future translational research.

Unlike mRNAs that are predominantly cytoplasmic, lncRNAs can be nuclear and/or cytoplasmic (*Cabili et al., 2015*). While nuclear lncRNAs are bona fide noncoding, cytoplasmic lncRNAs have the potential to be translated if they interact with ribosomes. Indeed, hundreds of cytoplasmic lncRNAs, harboring evolutionary conserved open-reading frames (ORFs) are associated with ribosomes (*Carlevaro-Fita et al., 2016*; *Ingolia et al., 2014*; *Ingolia et al., 2011*; *Hartford and Lal, 2020*), and small proteins derived from such mis-annotated lncRNAs have vital functions in many organisms ranging from bacteria to humans (*Makarewich and Olson, 2017*; *Storz et al., 2014*). These small proteins regulate diverse processes including stress response (*Slavoff et al., 2014*; *Guo et al., 2014*), development (*Chng et al., 2013*; *Kondo et al., 2010*), calcium homeostasis (*Anderson et al., 2015*), cardiac function (*Makarewich et al., 2018a*), metabolism (*Makarewich et al., 2018b*), myoblast fusion (*Nelson et al., 2016*), and mitochondrial respiration (*Stein et al., 2018*). Given the vast number of uncharacterized lncRNAs, of which a substantial proportion are bound to ribosomes, detailed characterization of lncRNAs on a case-by-case basis will significantly advance our understanding of their roles in normal development and diseases such as cancer.

Here, we report the initial characterization of a naturally occurring, highly conserved small protein encoded by *LINC00675*, a putative lncRNA that we found to be induced by the pioneer transcription factor FOXA1. We therefore termed this putative lncRNA as *FORCP* (*F*OXA1 *R*egulated *C*onserved Small *P*rotein). Our data identify *FORCP* as a novel, gastrointestinal (GI) tract-specific, protein-coding gene translated from a transcript annotated as a lncRNA. We show that endogenous FORCP plays a role in inducing apoptosis during endoplasmic reticulum (ER) stress and in the inhibition of proliferation and tumorigenicity in well-differentiated colorectal cancer (CRC) cells.

## Results

### FORCP is transcriptionally activated by FOXA1 in well-differentiated CRC cells

To identify lncRNAs that could function as tumor suppressors in CRC, we examined their expression in a CRC cohort. *FORCP* was one of the most significantly down-regulated lncRNAs in CRC tumors (*Figure 1A*). *FORCP* is transcribed from chromosome 17 and is antisense to *TMEM220-AS1* (*Figure 1—figure supplement 1A*). In normal human tissues, *FORCP* is expressed in a GI-tract-specific manner with high expression in the normal human colon and stomach (*Figure 1—figure supplement 1B*). In addition, in the colon *FORCP* was ~seven- and three-fold less abundant than the highly expressed lncRNAs *MALAT1* and *NORAD*, respectively (*Figure 1—figure supplement 1C*). Given the substantial downregulation of *FORCP* in CRC tumors and high expression in normal human colon tissue, we hypothesized that *FORCP* functions as a tumor suppressor in CRC.

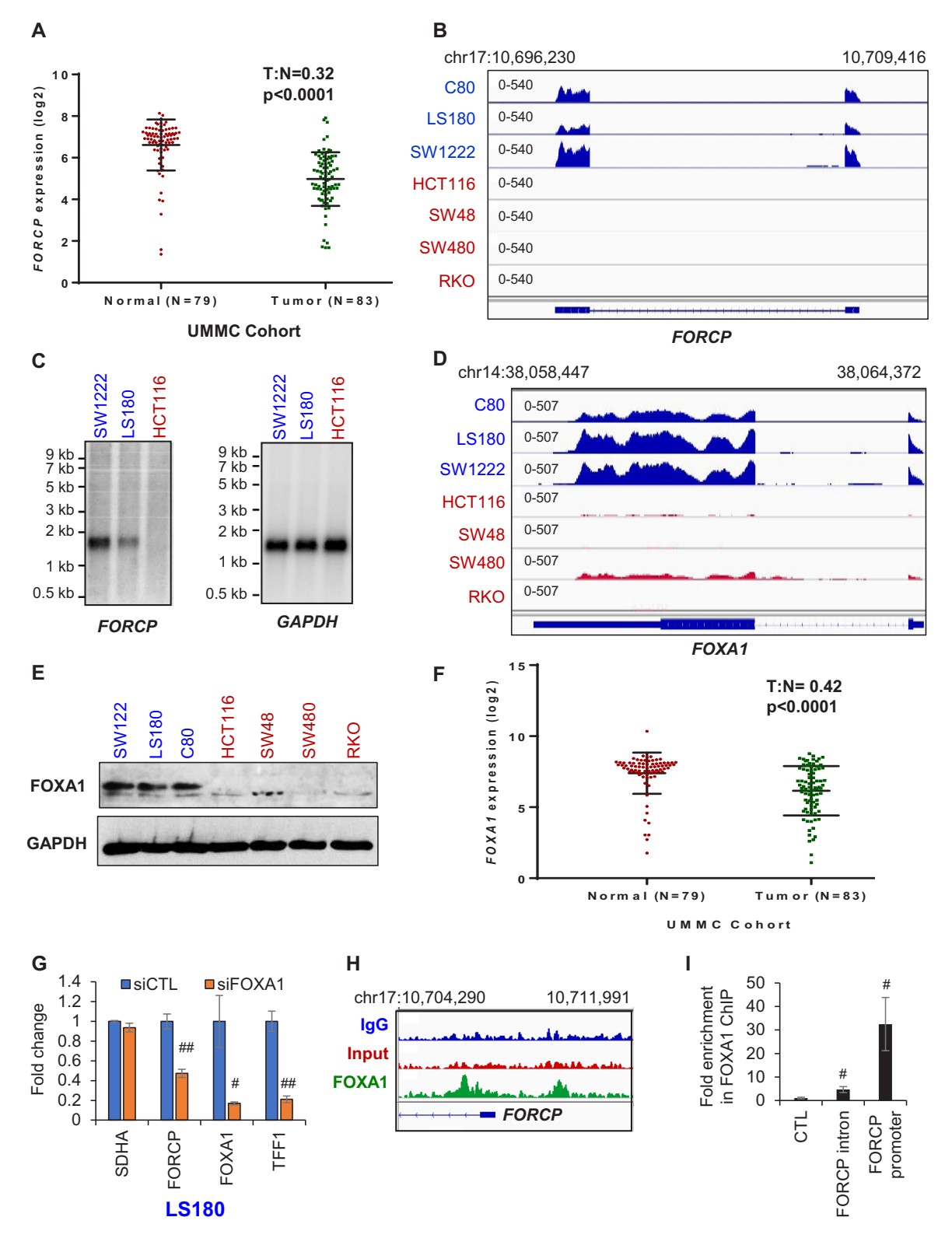

**Figure 1.** *FORCP* expression is restricted to well-differentiated CRC cells and is controlled by FOXA1. (**A**) Analysis of *FORCP* expression in CRC patient samples and matched normal colon in the UMMC cohort from which we performed lncRNA microarrays from 83 CRC patient samples and 79 matched normal tissue (Schetter et al., unpublished). T refers to tumors and N refers to normal human colon tissue. There were 79 and 83 samples for N and T, respectively. UMMC refers to University of Maryland Medical Center Cohort. (**B**) IGV snapshot from our RNA-seq shows robust *FORCP* expression in

*Figure 1 continued on next page*

*Figure 1 continued*

well-differentiated CRC cell lines (blue) and undetectable *FORCP* expression in poorly differentiated CRC lines (red). (C) Northern blot analysis was performed for *FORCP* RNA and the loading control *GAPDH* mRNA in well-differentiated (SW1222 and LS180) and poorly differentiated CRC cells (HCT116). (D, E) IGV snapshot from our RNA-seq (D) and immunoblotting (E) demonstrating higher expression of *FOXA1* in well-differentiated (blue) compared to poorly differentiated (red) CRC cell lines. *GAPDH* served as a loading control (E). (F) Decreased *FOXA1* expression in CRC tumor samples compared to normal samples in the UMMC cohort is shown. (G) qRT-PCR analysis following *FOXA1* knockdown in LS180 cells demonstrates efficient knockdown of *FOXA1*, and decreased *FORCP* and *TFF1* levels. qRT-PCR was normalized to *GAPDH*. *SDHA* served as a negative control. (H) IGV snapshot from FOXA1 ChIP-seq from LS180 cells shows two FOXA1 peaks located in the intronic and promoter region of *FORCP*, respectively. (I) Association of FOXA1 with the intronic and promoter region of *FORCP* was validated by ChIP-qPCR. Error bars in (G) and (I) represent standard deviation from three experiments. Error bars in panels G and I represent standard deviation (SD) from three experiments. #p<0.01, ##p<0.001.

The online version of this article includes the following source data and figure supplement(s) for figure 1:

Source data 1. FOXA1 and GAPDH immunoblots for *Figure 1E*.
Figure supplement 1. *FORCP* expression in cell lines and normal human tissues.
Figure supplement 2. Expression of specific gens in CRC cell lines.
Figure supplement 3. FOXA1 regulates *FORCP* expression.
Figure supplement 3—source data 1. FOXA1 and GAPDH immunoblots for *Figure 1—figure supplement 3D and E*.

Surprisingly, *FORCP* was almost undetectable in the commonly used CRC cell lines HCT116, RKO and SW48 and was expressed in only 3 out of 60 cell lines in the NCI-60 panel in previously published (*Reinhold et al., 2019*) RNA-seq data (*Figure 1—figure supplement 1D and E*, *Supplementary file 1* and *2*). However, in our RNA-seq from 7 CRC cell lines of which three were well-differentiated (C80, LS180, and SW1222) and four were poorly differentiated (HCT116, SW48, SW480, and RKO), *FORCP* was almost exclusively expressed in the well-differentiated CRC lines (~4400 fold change) (*Figure 1B* and *Supplementary file 2*). It should be noted that the RNA-seq snapshot (*Figure 1B*) does not include *TMEM220-AS1*, because it was not expressed in this 7 CRC cell line panel. The protein-coding genes *CEACAM5*, *CDX1*, *CDX2*, *KRT20*, and *VIL1*, known to be abundant in the normal human colon, showed robust expression only in the well-differentiated CRC cell lines and served as positive controls (*Figure 1—figure supplement 2A*). The annotated spliced *FORCP* transcript is ~1.5 kb long and has a canonical polyadenylation signal at its 3′ end (*Figure 1—figure supplement 2B*). Northern blotting showed that *FORCP* runs at the expected size in SW1222 and LS180 and is undetectable in HCT116 (*Figure 1C*).

To identify the transcription factor that controls *FORCP* expression, we analyzed ENCODE ChIP-seq data (chromatin immunoprecipitation followed by next-generation sequencing). We found ChIP-seq peaks for FOXA1, Glucocorticoid receptor (GR), GATA3 and MYC at the *FORCP* locus (*Figure 1—figure supplement 3A*). Among these, FOXA1 was the only transcription factor that was significantly more abundant at the mRNA (~16 fold, p<0.0001) and protein level in the well-differentiated lines as compared to the poorly differentiated lines (*Figure 1D,E*, *Figure 1—figure supplement 3B* and *Figure 1—source data 1*). Additionally, *FOXA1* was significantly down-regulated in CRC tumors (*Figure 1F*) and knockdown of *FOXA1* in LS180 and SW1222 significantly decreased *FORCP* levels (*Figure 1G* and *Figure 1—figure supplement 3C–E* and *Figure 1—figure supplement 3—source data 1*). By performing FOXA1 ChIP-seq (*Lazar et al., 2020*) followed by ChIP-qPCR, we identified and validated FOXA1 binding to the *FORCP* locus (*Figure 1H and I*). These data suggest that FOXA1 enhances *FORCP* transcription in the well-differentiated CRC cells.

## Discovery of an endogenous small protein encoded by the *FORCP* locus

The function of a lncRNA is often determined by its subcellular localization. RNA fluorescence in situ hybridization (RNA-FISH) and qRT-PCR from nuclear and cytoplasmic fractions suggested that *FORCP* is predominantly cytoplasmic, similar to *GAPDH* (*Figure 2A*, *Figure 2—figure supplement 1A and B*). Next, using PhyloCSF, an algorithm that determines the probability of a multi-species nucleotide sequence alignment representing a protein-coding region, we found that *FORCP* RNA has a positive maximum codon substitution frequency (CSF), similar to the protein-coding gene *GAPDH* but unlike the lncRNA *NEAT1* which had a negative maxCSF score (*Figure 2B*). Analysis of evolutionary conservation revealed a short, 240 nt ORF in *FORCP* (*Figure 2—figure supplement 1C*) that has the potential to be translated into a 79 amino acid small protein highly conserved between mammals (*Figure 2—figure supplement 2A*). Moreover, human *FORCP* and its mouse

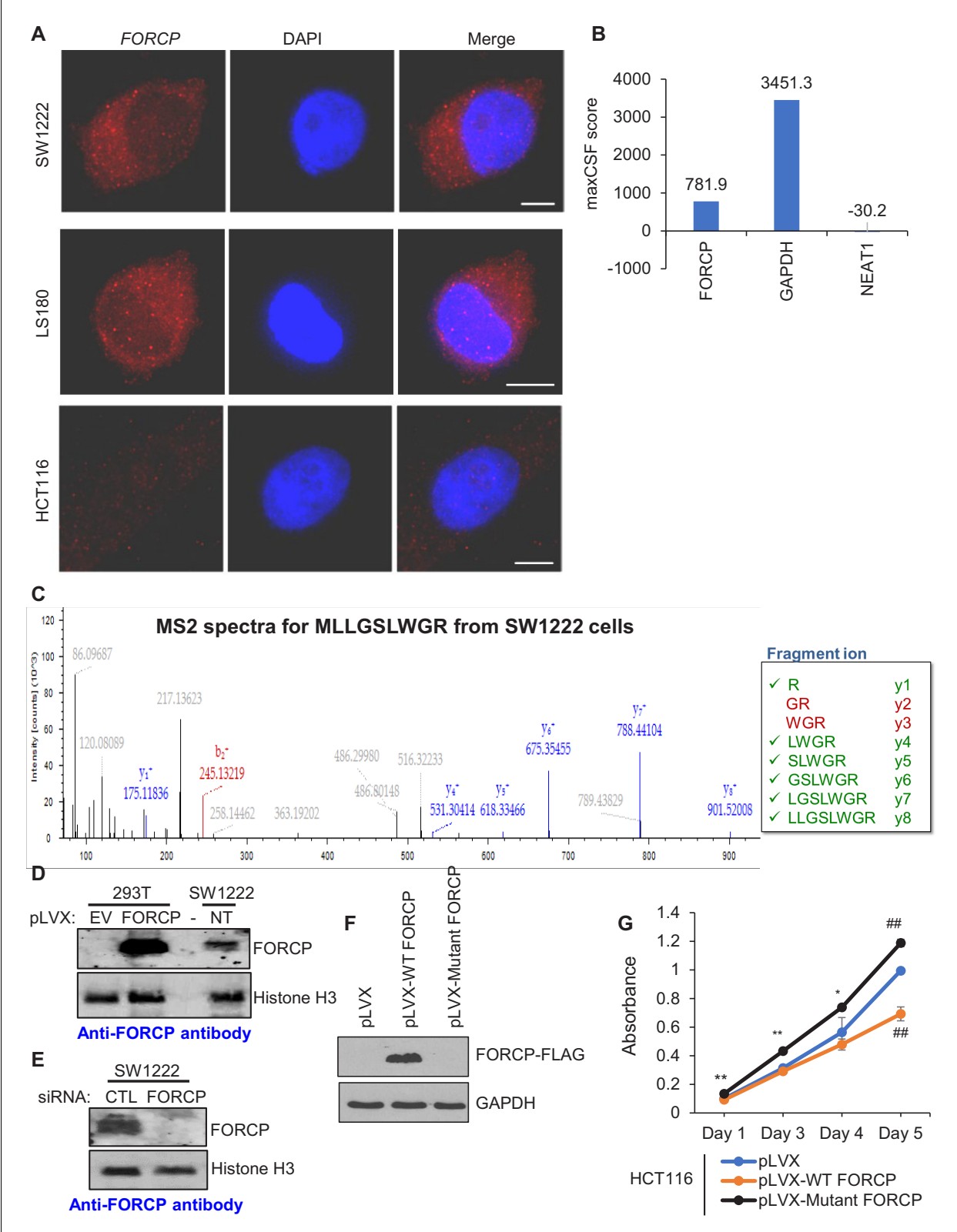

**Figure 2.** *FORCP* transcript is predominantly cytoplasmic and is translated to a naturally occurring 79 amino acid protein. (**A**) Single-molecule RNA-FISH was performed for *FORCP* from SW1222, LS180 and HCT116 cells. DNA was counterstained with DAPI. (**B**) Maximum CSF (maxCSF) scores for *FORCP*, *GAPDH* (protein-coding) and *NEAT1* (non-coding) transcripts were determined by PhyloCSF analysis. (**C**) Mass spectrometry from SW1222 tryptic digests identifies six out of eight fragment ions (y ions shown in green) from a N-terminal fragment of FORCP protein corresponding to the peptide

*Figure 2 continued on next page*

*Figure 2 continued*

sequence MLLGSLWGR. (**D, E**) Detection of overexpressed FORCP protein in 293 T cells and endogenous FORCP protein in SW1222 (NT) by immunoblotting using an anti-FORCP antibody. FORCP protein was not detected in 293 T cells transduced with the empty lentiviral expression vector (EV) or in SW1222 cells following *FORCP* knockdown with siRNAs. Histone H3 served as a loading control. (**F**) Detection of FORCP-FLAG in HCT116 cells transduced with empty vector (pLVX), pLVX-WT FORCP, or pLVX-Mutant FORCP by immunoblotting using anti-FLAG antibody. GAPDH served as loading control. (**G**) Cell viability assays were performed from HCT1116 cells transduced with empty vector (pLVX), pLVX-WT FORCP, or pLVX-Mutant FORCP. Error bars in panel H represent SD from three experiments. *p<0.05, **p<0.05, ##p<0.001.

The online version of this article includes the following source data and figure supplement(s) for figure 2:

**Source data 1.** FORCP and histone H3 immunoblots for *Figure 2D*.
**Source data 2.** FORCP and histone H3 immunoblots for *Figure 2E*.
**Source data 3.** FORCP-FLAG and GAPDH immunoblots for *Figure 2F*.
**Figure supplement 1.** *FORCP* RNA is predominantly cytoplasmic and harbors a short ORF.
**Figure supplement 2.** Mammalian conservation of FORCP.
**Figure supplement 3.** Overexpression experiments suggest that *FORCP* may not be a bifunctional gene.

homolog (ENSMUSG00000085683.2/*9130409J20Rik)* share several common features. First, they are located at a syntenic region on human chromosome 17 and mouse chromosome 11. Second, they have well-conserved 240 nt ORFs that have the potential to be translated into a 79 amino acid small protein (*Figure 2—figure supplement 2B*). Third, similar to the human *FORCP,* the mouse homolog shows biased expression in the GI-tract (*Figure 2—figure supplement 2C*).

To determine if the *FORCP* RNA is translated into an endogenously expressed protein, we performed mass spectrometry analysis from SW1222 whole cell lysates. We identified an N-terminal peptide MLLGSLWGR that was unique to the FORCP protein (*Figure 2C*). We therefore generated a rabbit polyclonal antibody against the FORCP protein. This antibody detected endogenous FORCP in SW1222 whole cell lysates that co-migrated with untagged FORCP protein overexpressed in 293T cells transduced with a lentivirus (pLVX-FORCP) (*Figure 2D* and *Figure 2—source data 1*). Knockdown of *FORCP* in SW1222 abolished expression of the FORCP protein (*Figure 2E* and *Figure 2—source data 2*). These data indicate that *FORCP* is a novel protein-coding gene and not a lncRNA. However, it could potentially act as a bi-functional gene like other lncRNAs (*Nam et al., 2016*; *Cai et al., 2017*; *Chooniedass-Kothari et al., 2004*). To test this, we overexpressed wild-type (WT) *FORCP* full-length RNA or a *FORCP* mutant in which the start codon ATG was mutated to ATT and inserted three copies of the FLAG-tag before the stop codon. As expected, the FORCP-FLAG protein was expressed only in the pLVX-WT *FORCP* transduced cells (*Figure 2F* and *Figure 2—source data 3*). At the RNA level, the expression levels of the WT and Mutant *FORCP* RNAs were comparable (*Figure 2—figure supplement 3A*). Overexpression of *FORCP*-WT but not the mutant *FORCP* in HCT116 resulted in significant growth defects as measured by cell proliferation assays (*Figure 2G*), colony formation assays on plastic and soft agar colony formation assays (*Figure 2—figure supplement 3B and C*) indicating that exogenous *FORCP* inhibits proliferation as a protein-coding gene.

## FORCP protein is localized to the endoplasmic reticulum

To examine the subcellular localization of FORCP protein, we overexpressed FORCP-GFP fusion protein in 293T cells and then verified the expression of this fusion protein by immunoblotting (*Figure 3—figure supplement 1A and B* and *Figure 3—figure supplement 1—source data 1*). Unlike GFP which was localized to the nucleus and cytoplasm, FORCP-GFP was predominantly localized to the ER, colocalizing with the ER marker PDI (protein disulfide isomerase) (*Figure 3A and B*). However, a small fraction of FORCP-GFP was also found in the perinuclear region. This localization pattern was also observed when we inserted three FLAG epitope tags in-frame before the stop codon of *FORCP* ORF within the full-length (FL) *FORCP* transcript (*Figure 3—figure supplement 1C*). Further analysis using the TMHMM Server v. 2.0 indicated that FORCP could be a double-pass transmembrane protein with two transmembrane helices and BLASTX analysis revealed that the C-terminus of the FORCP protein is homologous to TMEM238, a transmembrane protein (*Figure 3—figure supplement 1D and E*). To determine if the endogenous FORCP protein resides in the ER, we assayed for its localization using the anti-FORCP antibody. In LS180, we observed cytoplasmic ER-like staining of the FORCP protein that was strongly reduced upon knockdown of *FORCP* with

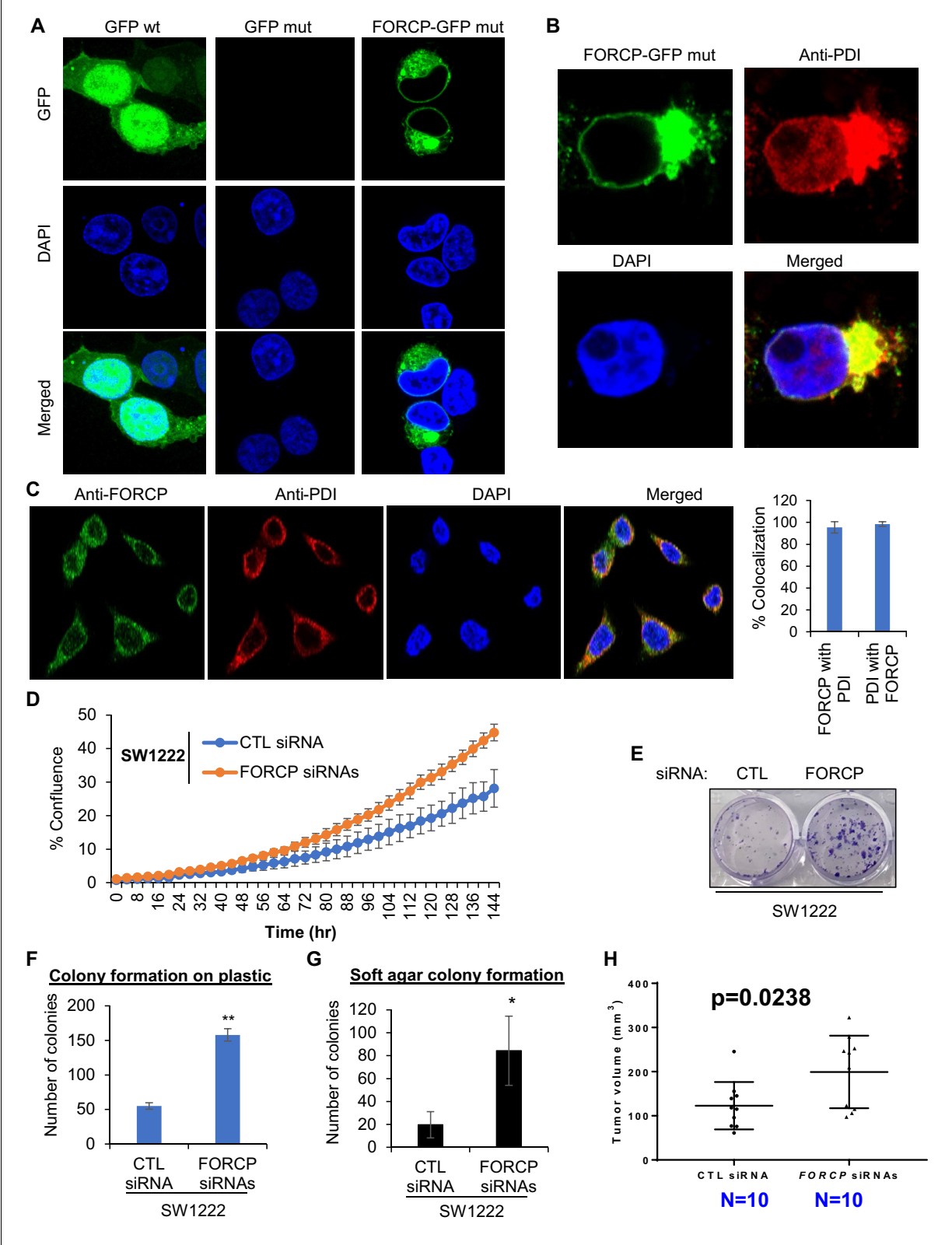

**Figure 3.** FORCP protein is localized to the endoplasmic reticulum and its knockdown leads to growth advantage. (A, B) Confocal microscopy was performed following transfection of 293 T cells with GFP wt, GFP with mutated ATG (GFP mut), or FORCP-GFP mut. Immunostaining for the ER marker PDI (B) shows colocalization of FORCP-GFP mut with PDI. DNA was counterstained with DAPI. (C) Confocal microscopy (left panel) following immunostaining of LS180 cells with anti-FORCP and anti-PDI antibodies shows colocalization of endogenous FORCP with the ER marker PDI. DNA was

*Figure 3 continued on next page*

*Figure 3 continued*

counterstained with DAPI. Colocalization of FORCP and PDI (right panel) was analyzed using ZEISS ZEN Desk microscope software from 100 individual cell images and converted to percentage co-localization. (D–H) The effect of *FORCP* knockdown in SW1222 cells on proliferation and tumorigenicity was assessed by Incucyte live cell proliferation assays (D), colony formation assays (E–G) and mouse xenograft experiments (H). Image from a representative colony formation on plastic experiment is shown in panel E and the data from three experiments is quantitated in panel F. For mouse xenograft experiments 'N' refers to the number of tumors 18 days after injecting the mice with SW1222 cells that were transfected for 48 hr with CTL or *FORCP* siRNAs. Error bars in panels D, F, and G represent SD from three experiments. $^*$p< 0.05, $^{**}$p< 0.005.

The online version of this article includes the following source data and figure supplement(s) for figure 3:

**Figure supplement 1.** Tagged FORCP protein is localized to the ER.
**Figure supplement 1—source data 1.** GFP and GAPDH immunoblots for *Figure 3—figure supplement 1B*.
**Figure supplement 2.** Immunostaining experiments show specificity of the FORCP antibody.

siRNAs (*Figure 3—figure supplement 2A*). Colocalization experiments with PDI suggested that endogenous FORCP is localized to the ER (*Figure 3C*).

## *FORCP* inhibits basal proliferation and induces apoptosis upon ER stress

To determine the function of *FORCP*, we knocked it down using siRNAs (*Figure 3—figure supplement 2B*). Knockdown of *FORCP* in SW1222 resulted in increased proliferation (*Figure 3D*), increased clonogenic potential on plastic (*Figure 3E and F*), and increased colony formation on soft agar, a measure of tumorigenicity in vitro (*Figure 3G*). In concordance with these data, we observed enhanced tumor growth in mouse xenografts upon *FORCP* knockdown in SW1222 (*Figure 3H*). These data suggest that *FORCP* functions to inhibit proliferation and could suppress tumorigenicity.

The ER is the major organelle responsible for protein folding, translocation, post-translational modification, protein assembly into oligomeric complexes, lipid and sterol biosynthesis and calcium homeostasis (*Brodsky and Skach, 2011*; *Braakman and Bulleid, 2011*). In response to exogenous or endogenous agents causing ER dysfunction and accumulation of unfolded proteins in the lumen, the ER attempts to reestablish normal function by triggering the unfolded protein response (*Walter and Ron, 2011*; *Preissler and Ron, 2019*; *Patil and Walter, 2001*; *Kaufman, 1999*; *Harding et al., 2002*). If the damage is prolonged or too severe, pathways initiated in the ER induce cell death. Since FORCP protein is primarily localized to the ER, we sought to determine if it plays a role in response to ER stress. When we measured *FORCP* levels upon treatment of LS180 with the ER-stress inducers dithiothreitol (DTT) or tunicamycin (TM), we found that *FORCP* was upregulated ~3–4 fold (*Figure 4—figure supplement 1A and B*). Interestingly, unlike basal *FORCP* expression which was controlled by FOXA1, upregulation of *FORCP* upon ER stress in FOXA1 knockdown cells was less marked, perhaps due to a decrease in basal *FORCP* levels (*Figure 4—figure supplement 1C*). Upon ER stress, silencing *FORCP* using siRNAs resulted in significantly better survival as compared to the control as assessed by cell viability and colony formation assays (*Figure 4—figure supplement 1D–F*). The improved cell viability upon *FORCP* knockdown during ER stress was due to a decrease in apoptosis as measured by the sub-G1 population in LS180 (*Figure 4A* and *Figure 4—figure supplement 1G*) and SW1222 (*Figure 4—figure supplement 2*). The pro-apoptotic effect of *FORCP* was further confirmed by immunoblotting for the apoptosis marker cleaved caspase-3 (*Figure 4B* and *Figure 4—source data 1*).

To further establish these functions of *FORCP*, we used the CRISPR/Cas9 technology targeting a single guide RNA (sgRNA) near the *FORCP* start codon to generate *FORCP*-WT (wild-type) and isogenic *FORCP*-KO (knockout) LS180 cells (*Figure 4—figure supplement 3A*). As compared to *FORCP*-WT cells, *FORCP*-KO cells showed significant reduction in *FORCP* expression (*Figure 4—figure supplement 3B*) and decreased apoptosis in response to ER stress (*Figure 4C and D*, and *Figure 4—source data 2*), further establishing a pro-apoptotic role of *FORCP* during ER stress. In addition, the *FORCP*-KO cells displayed increased basal proliferation and clonogenicity as compared to *FORCP*-WT cells, further supporting our data from the *FORCP* knockdown experiments (*Figure 4E* and *Figure 4—figure supplement 3C*). Finally, in response to another stress, namely glucose deprivation induced by treating the cells with 2-deoxy-D-glucose (2-DG), the *FORCP*-KO cells displayed a modest but significantly improved viability as compared to *FORCP*-WT cells (*Figure 4—figure supplement 3D*).

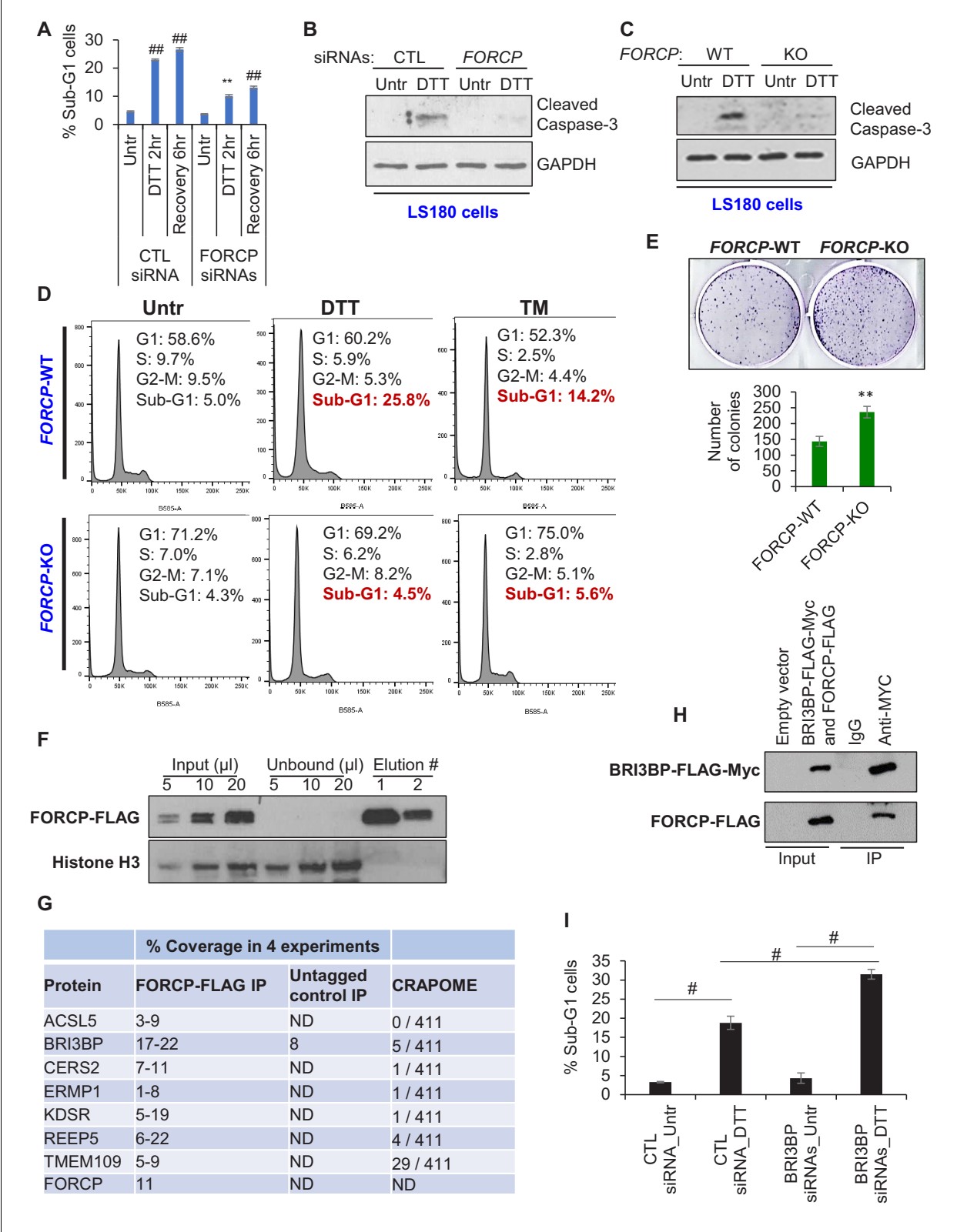

**Figure 4.** FORCP inhibits apoptosis in response to ER stress and interacts with BRI3BP. (**A**) PI staining and FACS analysis was performed from LS180 cells transfected for 48 hr with CTL siRNA or *FORCP* siRNAs and then left untreated (Untr) or treated with DTT for 2 hr or allowed to recover for 6 hr after DTT treatment. The Sub-G1 population from three independent experiments is shown in panel A. Analysis was performed using FlowJo software. (**B**) LS180 cells were transfected for 48 hr with CTL siRNA or *FORCP* siRNAs and then left untreated or treated with DTT for 2 hr followed by

*Figure 4 continued on next page*

*Figure 4 continued*

immunoblotting from whole cell lysates for the apoptosis marker cleaved caspase-3. GAPDH was used as loading control. (C) Immunoblotting for the apoptosis marker cleaved caspase-3 was performed from whole cell lysates prepared from *FORCP*-WT cells and the *FORCP*-KO LS180 cells untreated (Untr) or treated with DTT for 2 hr. GAPDH was used as loading control. (D) PI staining and FACS analysis was performed from *FORCP*-WT and isogenic *FORCP*-KO cells untreated (Untr) or treated with DTT for 2 hr or Tunicamycin (TM) for 6 hr. (E) Colony formation assays were performed 10 days after seeding *FORCP*-WT cells and the *FORCP*-KO cells in six-well plates. (F) HCT116 cells were transduced with empty vector (pLVX) or pLVX-FORCP-FLAG expressing FORCP-FLAG. Immunoblotting for FORCP-FLAG and Histone H3 was performed from lysates (input), unbound material (unbound) and the material from two consecutive elutions (#1 and #2). (G) Table shows the list of seven proteins that were identified in four biological replicates of anti-FLAG pulldowns followed by mass spectrometry from HCT116 cells transduced with pLVX-FORCP-FLAG (FORCP-FLAG IP) or empty vector (Untagged control IP). The range for percentage coverage for each protein in the four experiments is shown. The CRAPOME column shows the number of experiments in which these proteins were pulled down in 411 experiments listed in CRAPOME. 'ND' refers to undetermined. (H) 293 T cells were transfected with empty vector or co-transfected with constructs expressing BRI3BP-FLAG-Myc and FORCP-FLAG for 48 hr. Immunoblotting was performed using anti-FLAG antibody from whole cell lysates (Input) or material eluted following IP using IgG or anti-Myc antibody. (I) PI staining and FACS analysis was performed from LS180 cells transfected for 48 hr with CTL siRNA or *BRI3BP* siRNAs and then left untreated (Untr) or treated with DTT for 2 hr. Medium containing drug was removed and the cells were allowed to recover for 6 hr in fresh medium. The Sub-G1 population from three independent experiments is shown in panel I. Analysis was performed using FlowJo software. Error bars represent SD from three experiments. #p<0.01, **p<0.05, ##p<0.001. Figure legends for figure supplements.

The online version of this article includes the following source data and figure supplement(s) for figure 4:

**Source data 1.** Cleaved caspase-3 and GAPDH immunoblots for *Figure 4B*.
**Source data 2.** Cleaved caspase-3 and GAPDH immunoblots for *Figure 4C*.
**Figure supplement 1.** FORCP is upregulated upon ER stress to induce apoptosis.
**Figure supplement 2.** FORCP induces apoptosis upon ER stress in SW1222 cells.
**Figure supplement 3.** Targeted deletion of FORCP using CRISPR/Cas9.
**Figure supplement 4.** BRI3BP knockdown results in increased apoptosis upon ER stress.

## Identification of FORCP-interacting proteins

To gain insight into the molecular mechanism by which the FORCP protein functions, we decided to identify FORCP-interacting proteins. Using a lentivirus system, we overexpressed FORCP-FLAG in HCT116 cells and optimized the conditions for efficient pulldown of the FORCP-FLAG protein from whole cell extracts. As shown in the immunoblotting in *Figure 4F*, almost all of the FORCP-FLAG protein in the input was successfully pulled down. The abundant Histone H3 protein was not enriched in the pulldowns indicating the specificity of the pulldowns (*Figure 4F*). Having set up specific pulldown of the FORCP-FLAG protein, we performed pulldowns from four biological replicates followed by mass spectrometry. To reduce the number of false positives, we intersected the list of proteins enriched in our pulldowns with CRAPOME, a publicly available online tool that shows if a protein of interest is a common contaminant in more than 400 pulldown experiments. We found 16 FORCP-FLAG-interacting proteins enriched in all four pulldowns (*Supplementary file 3*). Of these 16 proteins, seven appeared in less than 30 of the 411 experiments in the CRAPOME dataset (*Figure 4G*). Of these seven proteins, we focused on BRI3BP (BRI3-binding protein) because it had been reported in a subnetwork of proteins important for ER biology (*Christianson et al., 2011*). We next validated the interaction between FORCP-FLAG and BRI3BP by co-transfecting 293 T cells with constructs expressing FORCP-FLAG and BRI3BP-FLAG-Myc followed by anti-Myc IP and immunoblotting using an anti-FLAG antibody (*Figure 4H*). To determine if BRI3BP has a function during ER stress, we knocked down endogenous *BRI3BP* in LS180 cells using siRNAs (*Figure 4—figure supplement 4A*) and measured apoptosis using PI staining and FACS analysis. Unlike *FORCP*, knockdown of *BRI3BP* resulted in significantly increased apoptosis when ER stress was induced using DTT (*Figure 4I* and *Figure 4—figure supplement 4B*). These data on the FORCP-BRI3BP interaction raise the possibility that FORCP could inhibit BRI3BP function or *vice versa* in the context of ER stress. However, this needs to be investigated in future studies. Together, our results indicate that *FORCP* is a novel protein-coding gene that was mis-annotated as a lncRNA. FORCP protein localizes to the ER and plays a role in suppressing basal proliferation and tumorigenicity, and inducing apoptosis during ER stress in well-differentiated CRC cells.

## Discussion

Here, we report the identification and initial characterization of FORCP as a novel, highly conserved, small protein expressed from a transcript annotated as a lncRNA (*LINC00675*). Our data demonstrate that FORCP is an ER-localized protein that is robustly expressed in well-differentiated CRC cells and that FOXA1 controls its transcription. Functionally, we found that *FORCP* inhibits proliferation at the basal level and is pro-apoptotic in the context of ER stress.

In contrast to current annotations and publications on this gene (*Li et al., 2015*; *Zeng et al., 2018*; *Zhong et al., 2018*; *Shan et al., 2018*; *Ma et al., 2018*; *Li et al., 2018*), we found that *FORCP* is mis-annotated as a lncRNA. Our discovery that *FORCP* is translated into a small protein contributes to the emerging notion that short ORFs are often hidden in lncRNAs and are translated into small proteins or micropeptides that play essential roles in regulation of fundamental biological processes in organisms ranging from bacteria to humans (*Hartford and Lal, 2020*; *Makarewich and Olson, 2017*; *Storz et al., 2014*; *Leslie, 2019*; *Orr et al., 2019*). Although the exact molecular mechanisms by which these small proteins mediate their effects remains largely unclear, there is some evidence that the small size of these proteins can allow them to block or boost the activity of the larger proteins they interact with (*Leslie, 2019*). For example, the micropeptide MOXI localizes to the inner mitochondrial membrane where it interacts with mitochondrial trifunctional protein (*Makarewich et al., 2018b*), whereas the micropeptide DWORF localizes to the sarcoplasmic reticulum in muscle cells where it enhances SERCA activity by displacing the SERCA inhibitors phospholamban, sarcolipin, and myoregulin (*Nelson et al., 2016*). In future studies, detailed investigation of the FORCP-BRI3BP interaction could help elucidate the precise mechanism(s) of its anti-proliferative functions and its role during ER stress. Because FORCP is localized to the ER, an organelle involved in the production, processing and transport of proteins and lipids in a cell, future studies that combine biochemical approaches with lipidomics and/or proteomics could help determine the exact molecular and cellular function(s) of FORCP. Given that we identified KDSR and CERS2 as FORCP-interacting proteins and that these proteins catalyze sequential steps in ceramide biosynthesis, it would be interesting to determine if FORCP has a role in ceramide biosynthesis. Additionally, determining the in vivo function of *FORCP* in mice could reveal a role of *FORCP* in normal colon biology and in colorectal cancer.

Our study emphasizes the need to identify the appropriate cell-type in which a lncRNA is robustly expressed. In the case of *FORCP*, we found that it is robustly expressed in normal human colon tissue and downregulated in CRC patients. Surprisingly, *FORCP* was not expressed in the commonly used CRC cell lines and in 57 cell lines in the NCI-60 panel. *FORCP* was abundant in well-differentiated CRC cells that are colon-like, consistent with high *FORCP* expression in the normal human colon tissue. Identifying the appropriate cell type was instrumental in the discovery of the FORCP protein, identification of FOXA1 as an upstream regulator of FORCP and for functional analysis. In summary, the discovery of *FORCP* as a novel protein-coding gene, together with recent reports on other lncRNA-encoded small proteins, reveals vital functions of small proteins translated from ORFs hidden in lncRNAs and underscores the need to determine the coding potential of cytoplasmic lncRNAs in an endogenous setting. Future studies will likely identify new mis-annotated lncRNAs that encode functional small proteins important for normal development and human diseases.

## Materials and methods

### Cell culture and siRNA transfections

HEK293T (293T), HCT116, SW48, SW480, RKO, C80, and LS180 cells were purchased from ATCC (Manassas, VA.). SW1222 cells were purchased from Millipore Sigma (St. Louis, MO). All cell lines were maintained in Dulbecco's Modified Eagle's (DMEM) (Thermo Fisher Scientific, Invitrogen) medium containing 10% (v/v) fetal bovine serum (Thermo Fisher Scientific) and 1% penicillin-streptomycin. Cells were cultured at 37°C, 5% $CO_2$. All cell lines were routinely checked for mycoplasma using the VenorGem Mycoplasma detection kit (Millipore Sigma-Aldrich). OnTarget siRNA SMARTpool for *FOXA1*, *FORCP* and *BRI3BP* were purchased from Thermo Fisher Scientific, Dharmacon. The Allstars Negative control (CTL) siRNAs were purchased from Qiagen. SiRNAs were reverse transfected at a final concentration of 20 nM, using Lipofectamine RNAiMAX (Thermo Fisher Scientific, Invitrogen) as instructed by the manufacturer.

## Cell lines

We confirm that the identity of all cell lines used in our study has been authenticated by STR profiling. All cell lines were routinely checked for mycoplasma.

## Ribo-zero paired-end RNA-Seq

Cells were cultured according to ATCC instruction and total RNA was isolated using RNeasy plus mini kit (Qiagen). Total RNA was fragmented, and the cleaved RNA fragments were copied into first strand cDNA using reverse transcriptase and random primers, followed by second strand cDNA synthesis using DNA polymerase I and RNase H. The resulting double-strand cDNA was used as the input to a standard Illumina library prep with end-repair, adapter ligation and PCR amplification being performed to generate a library for sequencing on HiSeq2000. The HiSeq Real Time Analysis software was used for processing the image files, the Illumina BCL2fastq1.8.4 was used for demultiplex and convert binary base calls and qualities to FASTQ format. Then, sequencing reads were trimmed for adapters and low-quality bases using Trimmomatic (version 0.3), and the trimmed reads were aligned to human hg19 reference genome and Ensembl annotation version 70 using TopHat_v2.0.8 software. Hundred bases long paired-end reads were examined for quality using PICARD and FastQC. The generated FASTQ files were mapped using TopHat2 alignment algorithm and differential gene expression analysis was performed using Cufflinks and Cuffdiff. Average read length was 110 nucleotides and there were ~150 million mapped reads per sample. For a non-zero-fold change, 0.01 was added to the FPKM of each gene.

The RNA-seq data and ChIP-seq data has been deposited to GEO. The series number is: GSE140536.

## Northern blotting

2 μg polyA+ RNAs were isolated using NucleoTrap mRNA Mini kit for polyA+ RNA extraction (Macherey-Nagel) and separated by 1% formaldehyde agarose gel. ssRNA Ladder (New England Biolabs) was used for marker. The agarose gel was prepared using NorthernMax Denaturing Gel Buffer (Ambion) and run using NorthernMax MOPS Gel Running Buffer (Ambion). RNA gel was washed two times with nucleotide-free water for 30 min each, followed by transfer in 10x SSC buffer to Amersham Hybond-N+ blot (GE Healthcare). RNA was then fixed by UV crosslinking with 120 mJ/cm$^2$. Labeling of random-primed probes was performed with the Prime-It II Random Primer Labeling Kit (Agilent) and a mammalian expression vector containing full-length *FORCP* cDNA. Hybridization was done overnight at 42˚C in ULTRAhyb hybridization buffer (Ambion) as per the manufacture's instructions. Blots were washed at 42˚C using 2X SSC+0.1% SDS and 0.1xSSC+0.1%SDS and imaged using a Phosphorimager.

## Cell viability and colony formation assays

For cell growth assay, SW1222 cells were reverse transfected with CTL-siRNA or FORCP-siRNA (20 nM) for 48 hr, then reseeded onto 48-well plates at 8000 cells per well and loaded into Incucyte live image system (ESSEN Bioscience). Images were taken every 4 hr for 6 days and cell growth was determined by Incucyte Analysis Software (ESSEN Bioscience). For cell viability assay, HCT116 cells stably expressing WT FORCP or Mutant FORCP were generated using Lentivirus package and transducing system. Cells were seeded onto 96-well plates at 1000 cells per well. Cell viability was determined using CCK-8 assay (Dojindo Laboratories). For colony formation on plastic, or soft agarose assays, cells were reseeded in a six-well plate at a density of 1000 cells per well. After 2 to 3 weeks, colonies were fixed with ice-cold 100% methanol for 5 min, stained with crystal violet and colonies were counted and analyzed using ImageJ. For SW1222 and LS180 cells, after transfection with CTL and *FORCP* siRNAs for 48 hr, cells were treated with or without ER stress agents, DTT (2 nM) or TM (2 μg/ml) for 2 hr, medium containing drug was removed and replaced with fresh medium. Cells were then seeded onto 96-well plates for cell viability assay or six-well plates for colony formation assays as described above.

## RNA isolation, qRT-PCR, and ChIP-qPCR

Total RNA from cell lines was extracted using RNeasy plus mini kit (Qiagen). For qRT-PCR analysis, 500 ng of total RNA was reverse transcribed using iScript Reverse Transcription kit (Bio-Rad), and

qPCR was performed using Fast SYBR Green Master Mix (Millipore Sigma) and StepOnePlus Real-time PCR system (ThermoFisher Scientific) according the manufacturer's instructions. Chromatin immunoprecipitation (ChIP) was performed using the Active Motif ChIP kit (Active Motif, Carlsbad, CA, USA) according to the manufacturer's instructions. Briefly, $5 \times 10^7$ LS180 cells grown in 15 cm plates were cross-linked with 1% formaldehyde, and cells were lysed and sonicated. Protein–DNA complexes were immunoprecipitated with control IgG or anti-FOXA1 (Santa Cruz) antibody. The IP material was washed and heated at 65°C overnight to reverse the crosslinks. ChIP DNA was column purified (Qiagen) and analyzed by qPCR.

Primer sequences for qRT-PCR and ChIP-qPCR are listed in *Supplementary file 3*.

## Fluorescence RNA in situ hybridization (RNA-FISH)

Thirty smFISH probes, spanning and antisense to *FORCP* transcript were designed using Stellaris Probe Designer and ordered from Biosearch Technologies (http://www.biosearchtech.com). Each probe was 20 nt long and its 3' end was modified with mdC (TEG-Amino). RNA-FISH was performed as follows: Cells were rinsed with 1X PBS and fixed by freshly made fixative solution (3:1 Methanol-Glacial Acetic Acid) for 10 min at room temperature. Freshly fixed cells were washed with washing buffer (10% formamide, 2XSSC) for 5 min and incubated in hybridization buffer (10% formamide, 2XSSC, 10% dextran sulfate) containing 125 nM of the probe in a humidified chamber in the dark for 16 hr at 37°C. After hybridization, cells were washed twice, 30 min each at 37°C, in washing buffer. DNA was counterstained with DAPI during the second wash. The coverslips were then washed with 4XSSC for 5 min at room temperature and mounted onto microscope slides with VectaShield Anti-fade Mounting Medium (Vector Laboratories; cat# H-1000). Z-stack images were acquired using Del-taVision microscope (GE) equipped with 60X/1.42 NA oil immersion objective (Olympus) and a CoolSNAP-HQ2 camera and processed through deconvolution and maximum intensity projection.

## Immunoblotting and subcellular fractionation

Total cell lysate was prepared using radioimmunoprecipitation (RIPA) buffer containing protease inhibitor cocktail (Roche) as previously described and protein concentration was determined using the Bicinchoninic Acid protein quantitation (BCA) kit (Thermo Scientific) (*Jones et al., 2015*). Subcellular fractionation followed by qRT-PCR for nuclear and cytoplasmic fractions was performed as previously described (*Li et al., 2017*). For immunoblotting, 10 μg whole cell lysate per lane was loaded onto a 12% SDS-PAGE gel and transferred to nitrocellulose membrane (Thermo Scientific). The following antibodies were used: anti-FLAG (1:2000, Sigma), anti-FOXA1 (1:1000, Santa cruz, USA), anti-Histone H3 (1:1000), anti-GFP (1:1000), anti-cleaved Caspase3 (1:1000), anti-GAPDH (1:3000) from Cell Signaling.

## Generation of anti-FORCP antibody

Rabbit polyclonal anti-FORCP antibody was generated by Abgent, USA, against the C-terminal peptide NH2-CYSLNIEVSPEKLDL-COOH that was used as antigen. The cysteine at the N-terminus of this peptide was for conjugation to KLH. Two New Zealand rabbits were immunized with KLH-conjugated peptide. ELISA assays showed the anti-sera was positive (1:4000 OD >1 at 450 nm), and 11.625 mg antibodies were produced. The immune sera were then affinity-purified using the immunizing peptide. The anti-FORCP antibody was used at a dilution of 1:500 for immunoblotting and 1:100 for immunostaining.

## Comparative analysis and evaluation of coding potential with PhyloCSF

Genomic coordinates for all exons of human *LINC00675* and *NEAT1*, *GAPDH* genes were downloaded from the UCSC Genome Browser (GRCh37/hg19) in BED format. A Multiz alignment of 46 vertebrates aligned to GRCh37/hg19 was downloaded separately for each gene, based on the extracted coordinates for mature transcript accordingly to the UCSC annotation and uploaded to Galaxy (https://usegalaxy.org/). PhyloCSF was applied to generated FASTA alignment for assessing the coding potential (the Codon Substitution Frequencies score - CSF) of each mature transcript and individual exon of analyzed genes as described in *Chaudhary et al., 2017*; (*Prensner and Chinnaiyan, 2011*). BLASTX (https://blast.ncbi.nlm.nih.gov/Blast.cgi) and CD-search tool (https://www.ncbi.nlm.nih.gov/ Structure/bwrpsb/bwrpsb.cgi) as well as CDD and pfam databases were used for

comparative analysis of all possible reading frames and estimation of potential to encode any recognizable protein domains. Multiple alignments for promoter regions and complete *LINC00675* mature transcripts were built using the Muscle program with default parameters. Genome rearrangements and comparisons between human and mouse genomes were analyzed using the Owen program for pair-wise alignments.

## Mass spectrometry from whole cell lysates

SW1222 or HCT116 cells were grown to 80% confluency in 10 cm plates, washed with PBS, scraped into 1 ml PBS, followed by centrifugation at 1000 g for 5 min at 4°C. One ml of urea buffer (50 mM HEPES, pH 8.0 and 8 M urea) was applied to each of the cell pellets. Cell lysates were sonicated on ice with three 5 s pulses (30 s pause between pulses). Lysates were cleared of cell debris by centrifugation at 14,000 rpm for 10 min at 4°C. Protein concentration was estimated using standard BCA assay. One hundred micrograms of lysate were reduced using dithiothreitol (Thermo Fisher) and alkylated with iodoacetamide (Thermo Fisher). Lysates were diluted to a final concentration of 2 M urea using 50 mM HEPES, pH 8.0 and proteins digested at 37°C overnight with 2 µg trypsin (Promega). Digestion was stopped by acidification of the extract using trifluoracetic acid and samples were vacuum centrifuged to dry. The first-dimension separation of the peptides was performed off line, using Waters Acquity HUPLC system with a fluorescence detector (Waters, Milford, MA) coupled to a 150 mm x 3.0 mm XBridge Peptide BEM 2. 5 µm C18 column (Waters, MA) operating at 0.35 ml/min. The dried peptides were reconstituted in 100 µl of mobile phase A (3 mM ammonium bicarbonate, pH 8.0). Mobile phase B was 100% acetonitrile (Thermo Fisher). The column was washed with mobile phase A for 10 min followed by gradient elution 0–50% B (10–60 min) and 50–75% B (60–75 min) and fraction collected every minute. The fractions, collected from minute 10 to 75, were consolidated into 24 pools, dried in SpeedVac and stored at −80°C until analysis by mass spectrometry. The dried peptide pools were reconstituted in 0.1% TFA and subjected to nanoflow liquid chromatography (200 nl/min) (Thermo Easy nLC 1000, Thermo Scientific) coupled to high-resolution tandem MS (Q Exactive, HF, Thermo Scientific). Data depended acquisition of top 20 ions was done at resolution of 60,000 for MS1 with the automatic gain control (AGC) set at $3e^6$ over a mass range of 380–1580 m/z, followed by MS/MS analysis at a resolution of 15,000 with AGC set at $2e^5$. Precursor ion isolation width was set at 1.4 m/z and normalized collision energy at 27, and charge state one and unassigned charge states were excluded. Acquired MS/MS spectra were searched against a FASTA file containing all potential FORCP tryptic peptides, along with a contaminant protein database, using SEQUEST and percolator at 0.01% FDR in Proteome Discoverer 2.2 software (Thermo Scientific, CA). The precursor ion tolerance was set at 10 ppm and the fragment ions tolerance was set at 0.02 Da along with methionine oxidation included as dynamic modification. Only fully tryptic peptides with up to two mis-cleavages were considered.

## Constructs, plasmid transfection, and immunostaining

pEGFP-FORCP constructs were generated by cloning the ORF of FORCP-WT or mutant ATG of ORF into pEGFP-N1 vector with Age1/Not1 sites. For pLVX-FORCP 3xFLAG constructs, full length of FORCP-WT or FORCP-mut containing 3xFLAG in its C-terminus was subcloned into lentivirus vector pLVX-PURO with EcoR1/Xba1 sites. The BRI3BP-FLAG-Myc construct was purchased from OriGene technology, USA.

For transfection and immunostaining, 293 T cells were seeded at 300,000 cells per well in a six-well plate. After 24 hr, the cells were transfected using Lipofectamine 2000 (life technology Invitrogen) according the manufacturer's instruction. Forty-eight hours after transfection, cells were reseeded onto chamber slides (Thermo Fisher Scientific, USA) and fixed with 4% paraformaldehyde for 10 min at room temperature (RT). Fixed cells were permeabilized by 0.5% Triton X-100 for 10 min at RT and stained with primary antibodies anti-FLAG M2 (Cell Signaling, rabbit), PDI (Sigma, mouse) or anti-FORCP custom antibody (Abgent) overnight at 4°C. After washing the cells three times with PBS, secondary antibody was added and incubated at RT for 1 hr. DNA was stained with DAPI (blue). Images were taken using a confocal microscope (Zeiss LSM 880 NLO Airyscan).

## Lentivirus particle package and transduction

293 T cells were seeded onto six-well plates. pLVX vector, pLVX-FORCP WT or pLVX-FORCP Mut were transfected with lentivirus package vectors using lipofectamine 2000 (Life Technologies Invitrogen) as directed by the manufacturer. Virus was collected after 48 hr and 72 hr post-transfection. Virus titer was determined by serial dilution method, and MOI equal to one was used for transducing HCT116 cells. Stable transduced cells were generated by puromycin selection.

## Flow cytometry cell cycle assays

LS180 or SW1222 cells were reverse transfected with CTL siRNA, *FORCP* siRNAs or *BRI3BP* siRNAs for 48 hr. Cells were treated with or without ER stress agent DTT (2 nM) for 2 hr. For recovery experiments, medium containing the drug was removed and refreshed with new culture medium. Cells were fixed with ice-cold 75% ethanol for 24 hr and stained with propidium iodide (Sigma) in the presence of RNase A (Qiagen). DNA content was analyzed on a FACSCalibur flow cytometer (BD Biosciences) and data analyzed using FlowJo software.

## Mouse xenograft assays

Animal protocols (protocol number LC-070–3) were approved by the National Cancer Institute Animal Care and Use Committee following AALAAC guidelines and policies. SW1222 cells were transfected with CTL siRNA or *FORCP* siRNAs for 48 hr. Cells were then trypsinized and washed with PBS. Live cells were counted with trypan blue exclusion and $1 \times 10^6$ cells were mixed with 30% Matrigel in PBS on ice and injected into the flanks of 6- to 8-week-old female athymic nude mice (Animal Production Program, Frederick, MD, USA) (each group N = 10). Tumor volume was measured twice a week after 1 week of injection. Four weeks after inoculation, mice were terminated according AALAAC protocol.

## CRISPR/Cas9-mediated targeted deletion of FORCP

CRISPR/Cas9 mediated *FORCP* knockout LS180 cells were generated by Synthego Corporation (Redwood City, CA, USA). To generate KO cells, Ribonucleoproteins containing the Cas9 protein and synthetic chemically modified sgRNA were electroporated into LS180 cells using Synthego's optimized protocol. Editing efficiency was assessed 48 hr post-electroporation by extracting genomic DNA from a pool of transfected cells followed by PCR amplification and Sanger sequencing. The resulting chromatograms were processed using Synthego Inference of CRISPR edits software (ice. synthego.com). The pool of cells was then seeded in 96-well plates at one cell per well followed by clonal selection and RT-qPCR to determine the effect on FORCP mRNA levels.

For indel analysis using Illumina sequencing, 20 ng of gDNA was used as template to amplify around the sgRNA target site using the primers listed. Briefly, two rounds of PCR were performed using the Kapa HiFi 2X mastermix in order to generate amplicons that can be sequenced using the Illumina MiSeq 2 × 150 format. Paired-end reads were generated, merged using FLASH (*Magoč and Salzberg, 2011*), filtered for quality, and subsequently mapped to the reference amplicon sequence using bwa mem. Sorted and indexed BAM files, generated by samtools, were then visualized using the Integrative Genomics Viewer (IGV).

## Immunoprecipitation followed by mass spectrometry

HCT116 cells were transduced with pLVX or pLVX-FORCP 3xFLAG using a lentiviral package system (in 293 T cells) and then selected with Puromycin. For each IP reaction, 200 µl of protein A/G magnetic beads (Thermo Scientific Cat#88803) were coated with 20 µg of M2 Flag antibody (Sigma, Cat# F3165) in 500 µl lysis buffer (20 mM Tris-HCl pH 7.5, 100 mM KCl, 5 mM MgCl$_2$ and 0.3% NP40) and then incubated at 4℃ with rotation overnight. The antibody-coated beads were captured using a magnetic stand and lysis buffer was removed. Beads were then washed four times using 1 ml PBS per wash. For lysate preparation, twenty 150 mm (p150 plate) plates of pLVX or pLVX-FORCP-FLAG expressing cells were used for each IP reaction. For each p150 plate, 1 ml of lysis buffer containing protease inhibitor cocktail was added directly to the plate. Then, the cells were scraped using a rubber policeman and combined the material from the 20 plates into a 50-ml falcon tube. Cell lysate was collected after centrifugation at 10,000 x g for 10 min at 4℃. 20 ml of lysate then was added to anti-Flag antibody coated beads and incubated at 4℃ for 4 hr with rotation. The IP

material was washed with 20 ml of lysis buffer for four times. After the final wash, 150 µl of 2xSDS loading buffer was added and incubated at 95℃ for 5 min to elute IP material from the beads. Four replicates of IP material were then subjected to mass spectrometry.

For mass spectrometry from the IP material, the eluted proteins from the above IPs were fractionated by SDS-PAGE and in-gel digested with trypsin, as described (*Shevchenko et al., 2006*). Resultant peptides were analyzed on a Thermo Orbitrap Fusion mass spectrometer with parent full-scan mass spectra collected in the Orbitrap mass analyzer set to acquire data at 120,000 FWHM resolution and HCD fragment ions detected in the ion trap. Proteome Discoverer 2.2 (Thermo) was used to search the data against human proteins from the UniProt database using SequestHT. The Percolator node was used to score and rank peptide matches using a 1% false discovery rate.

## Acknowledgements

We thank Drs. Glenn Merlino (NCI, NIH), Tom Misteli (NCI, NIH) and Gisela Storz (NICHD, NIH) for their comments on the manuscript. We thank the Genomics Core, the Flow Cytometry Core and the Microscopy Core Facility of the Center for Cancer Research (CCR) of the National Cancer Institute (NCI), NIH for their service. This research was supported by the Intramural Research Program (AL, MIA, SA, TA, CCH, LMJ, BT, PSH, RC, and PSM) of the National Cancer Institute (NCI), Center for Cancer Research (CCR), NIH, by the Intramural Research Program of the National Library of Medicine, NIH (SAS) and by the Intramural Research Program of the National Institute on Aging, NIH (MG). KVP lab was supported by grants from NSF-EAGER [1723008], Cancer center at Illinois seed grant, Prairie Dragon Paddlers support, NIH R21 AG065748 and NIH R01 GM132458.

## Additional information

### Competing interests

Ashish Lal: Reviewing editor, *eLife*. Sudipto Das, Thorkell Andresson: is affiliated with Leidos Biomedical Research, Inc. The other authors declare that no competing interests exist.

### Funding

| Funder | Grant reference number | Author |
| --- | --- | --- |
| National Institutes of Health | ZIA BC 011646 | Ashish Lal |
| National Science Foundation | NSF-EAGER | Kannanganattu V Prasanth |
| National Institutes of Health | R01 GM132458 | Kannanganattu V Prasanth |
| National Cancer Institute | Intramural Research Program | Corrine CR Hartford<br>BinWu Tang<br>Lisa M Jenkins<br>Raj Chari<br>S Perwez Hussain<br>Paul S Meltzer<br>Mirit I Aladjem<br>Thorkell Andresson |
| National Institutes of Health | Intramural Research Program of the National Institute on Aging | Stefan Ambs |
| National Institutes of Health | | Myriam Gorospe |
| Cancer Center at Illinois | Seed Grant | Kannanganattu V Prasanth |
| National Institutes of Health | Intramural Research Program of the National Institute on Aging | Myriam Gorospe |
| National Institutes of Health | R21AG065748 | Kannanganattu V Prasanth |

The funders had no role in study design, data collection and interpretation, or the decision to submit the work for publication.

## Author contributions
Xiao Ling Li, Raj Chari, Data curation, Methodology; Lőrinc Pongor, BinWu Tang, Ana I Robles, Svet-lana A Shabalina, Formal analysis; Wei Tang, Sudipto Das, Bruna R Muys, Sarah B Lazar, Aaron Schetter, Lisa M Jenkins, Data curation; Matthew F Jones, Qinyu Hao, Qinyu Sun, Jennifer L Martin-dale, Robert L Walker, Stefan Ambs, Myriam Gorospe, Methodology; Emily A Dangelmaier, Corrine CR Hartford, S Perwez Hussain, Investigation; Ioannis Grammatikakis, Investigation, Writing - review and editing; Curtis C Harris, Paul S Meltzer, Resources; Kannanganattu V Prasanth, Resources, Inves-tigation, Methodology; Mirit I Aladjem, Resources, Methodology; Thorkell Andresson, Data curation, Formal analysis, Methodology; Ashish Lal, Conceptualization, Supervision, Funding acquisition

## Author ORCIDs
Lőrinc Pongor https://orcid.org/0000-0001-5917-4628
Emily A Dangelmaier https://orcid.org/0000-0002-4698-2500
Ioannis Grammatikakis https://orcid.org/0000-0002-8455-1584
Qinyu Hao http://orcid.org/0000-0002-7059-7741
Kannanganattu V Prasanth http://orcid.org/0000-0003-4587-8362
Ashish Lal https://orcid.org/0000-0002-4299-8177

## Ethics
Animal experimentation: Animal protocols (protocol number LC-070-3) were approved by the National Cancer Institute Animal Care and Use Committee following AALAAC guidelines and policies.

## Decision letter and Author response
Decision letter https://doi.org/10.7554/eLife.53734.sa1
Author response https://doi.org/10.7554/eLife.53734.sa2

# Additional files

## Supplementary files
• Supplementary file 1. *FORCP*, *NORAD* and *MALAT1* expression from previously published RNA-seq data (*Reinhold et al., 2019*) from the NCI-60 panel of cell lines is shown.

• Supplementary file 2. RNA-seq was performed from 7 CRC lines. Poorly differentiated CRC lines are shown in yellow. Well-differentiated CRC lines are shown in blue. Data for *FORCP* (*LINC00675*) is shown in green.

• Supplementary file 3. Sequences of primers, gRNAs and siRNAs used in this study.

• Supplementary file 4. List of proteins that were found to interact with FORCP-FLAG in IPs followed by mass spectrometry.

• Transparent reporting form

## Data availability
RNA-seq and ChIP-seq data related to this manuscript has been submitted to GEO under accession number GSE140536.

The following dataset was generated:

| Author(s) | Year | Dataset title | Dataset URL | Database and Identifier |
|---|---|---|---|---|
| Pongor L | 2020 | A small protein encoded by a putative long noncoding RNA inhibits proliferation and tumorigenicity in human colorectal cancer cells | https://www.ncbi.nlm.nih.gov/geo/query/acc.cgi?acc=GSE140536 | NCBI Gene Expression Omnibus, GSE140536 |

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
