## [Decision Letter]

**Acceptance summary:**

This manuscript shows that a previously annotated long non-coding RNA expressed specifically in the GI tract actually encodes a small protein which plays a role in the ER stress pathway, regulating apoptosis and tumorigenicity.

**Decision letter after peer review:**

Thank you for sending your article entitled "A small protein encoded by a putative lncRNA regulates apoptosis and tumorigenicity in human colorectal cancer cells" for peer review at *eLife*. Your article is being evaluated by three peer reviewers, and the evaluation is being overseen by Maureen Murphy as the Senior and Reviewing Editor.

Summary:

This manuscript from the Lal lab describes the identification of a new micropeptide which they term FORCP encoded by an erroneously categorized lncRNA that is expressed in well-differentiated colon cancer cell lines. Using a combination of elegant bioinformatics and biochemical approaches, they report that the FORCP locus encodes for two exons and the spliced mRNA guides the expression of a small protein which can be detected by mass spectrometry and immunofluorescence. They localize this protein to the membrane of the ER and demonstrate that its expression can be induced by pharmacologically-induced ER stress. Using loss-of-function studies they describe tumor-suppressive effects of the encoded protein in in vitro and in vivo studies. The manuscript is well written and the results, for the most part, support the conclusions.

Essential revisions:

1) The concept that some (or many) lncRNAs actually encode small proteins is not new. As such the manuscript requires a much deeper mechanistic understanding of FORCP. Along these lines:

i) Identification of FORCP interacting proteins might provide allow the authors to elucidate a molecular function for this peptide, which would significantly increase the impact of this work.

ii) Alternatively, there is little insight here as to how FORCP functions in ER stress; the authors show that silencing FORCP leads to decreased induction of Bip and Chop, but this could be because silencing FORCP enhances cell death and these mRNAs are decreased. The manuscript ends with a bit of a disappointing note, and no hint as to the mechanism whereby FORCP works, in ER stress or in tumorigenesis or otherwise. Including clear data as to the mechanism whereby FORCP functions in ER stress or tumorigenesis would strengthen the impact of the work.

2) The most important technical weakness of this paper is the use of only a single siRNA pool for all the FORCP phenotypic experiments. Particularly because very general phenotypes are being studied (proliferation, cell death), it is essential that the authors do a better job of establishing the specificity of these results. Ideally, CRISPR would be used to knockout FORCP to confirm these findings. Although it may be difficult to grow the well-differentiated CRC cell lines as single cell clones, CRISPR knockouts can be highly effective in lentivirally-infected pools, so this is not a sufficient reason to avoid genome-editing experiments. Alternatively, it might be acceptable to use additional siRNAs and perform rescue experiments, but use of CRISPR would be more convincing. The authors would need to repeat all the experiments in Figures 6-7 with this additional loss-of-function approach.

3) The in vivo tumor growth data are lacking a clear description and appear to have been performed in non-conventional manner. Injecting 1 million cells embedded in Matrigel in the flank and reporting one volume measurement after 3 weeks does not provide sufficient detail into the effects of the studied protein. These studies must be performed and prevented more conventionally. In flank tumor models, tumor growth can be extended up to 1,000-1,500 cc and measurements on a daily or every other day should be reported. It is possible that the tumors did not grow at all, or grew very little as the data is presented. More importantly, there are no correlative studies to test the main findings in vitro, e.g., staining for Ki-67 or EdU incorporation (proliferation) and cleaved caspase-3/TUNEL (apoptosis) in the tumors. The inhibition of FORCP before injection of the tumors could very well have affected angiogenesis and not proliferation or resistance to apoptosis.

4) Mechanistically, the authors show that in vitro, FOXA1 binds to the promoter and an intronic region of FORCP gene. However, it is unclear if FOXA1 is required for the ER stress-dependent induction of FORCP.

5) In terms of ER stress, only two acute and non-physiological stresses, DTT and Tunicamycin were used. Since the authors suggest a tumor-suppressive role of FORCP, they should try more relevant stresses such as hypoxia and amino acid or glucose deprivation.

[Editors' note: further revisions were suggested prior to acceptance, as described below.]

Thank you for resubmitting your article "A small protein encoded by a putative lncRNA regulates apoptosis and tumorigenicity in human colorectal cancer cells" for consideration by *eLife*. Your revised article has been reviewed by three peer reviewers, and the evaluation has been overseen by a Reviewing Editor and Maureen Murphy as the Senior Editor. The following individual involved in review of your submission has agreed to reveal their identity: Constantinos Koumenis (Reviewer #2).

The reviewers have discussed the reviews with one another and the Reviewing Editor has drafted this decision to help you prepare a revised submission.

Summary:

This manuscript convincingly demonstrates that LINC00675 encodes a conserved small protein, which the authors term FORCP, that is expressed in the GI tract and in well-differentiated CRC cell lines. FORCP localizes to the ER, slows proliferation of CRC cell lines in vitro, and promotes apoptosis after ER stress.

The revised manuscript has been reviewed by all three reviewers. All feel that the absence of functional information about the role of FORCP in ER stress is a weakness. The reviewers appear willing to consider this further if the following changes are made in a further revised manuscript.

Essential revisions:

1) I do not think that the polysome fractionation data in Figure 2B is interpretable without the A260 absorbance data, which is necessary to determine which fractions contain polysomes. As the data are presented now, this is just a sucrose gradient fractionation, showing that FORCP RNA associates with fractions in the middle of the gradient. FORCP could be polysome associated, or it could be associated with some other EDTA-sensitive complex. Given the weakness of these data, and the authors' inability to perform the appropriate analysis due to the pandemic, I suggest that these data be removed. There are sufficient other data to show that FORCP encodes a protein without this experiment.

2) The images in Figure 3C, which show endogenous FORCP protein, do not very convincingly show co-localization with PDI (there is clearly some co-localization, but the FORCP distribution appears to be broader). Quantification of co-localization in a larger number of cells should be provided.

3) The inclusion of additional, more physiological stressors in the experiments described in Figure 4 is a positive development. However, the use of CoCl_2_ in lieu of "true hypoxia" (i.e., elicited by use of an environmental hypoxia chamber), is problematic. CoCl_2_ "mimics hypoxia" only as far as induction of HIF-1a or HIF-2a is concerned, via inhibition of the activity of prolyl-hydroxylases and post-translational stabilization of those proteins. Induction of ER stress and the UPR by hypoxia depends on actual reduction in intracellular O_2_ levels leading to accumulation of unfolded proteins. Therefore, Figure 4—figure supplement 3C should be removed and only the low glucose data be shown. A more detailed analysis on the role of FORCP in cell survival under hypoxia should be re-examined in more detail later.

4) The authors describe a proteomic study that identifies a protein associated with ER stress, BRI3BP. At the very least, presenting the IP-western indicating a functional interaction would provide some hint as to the mechanism whereby FORCP functions in ER stress; the authors would only have to present the proteomics and interaction and could save the remaining new data for a more comprehensive functional study.

---

## [Author Response]

Essential revisions:1) The concept that some (or many) lncRNAs actually encode small proteins is not new. As such the manuscript requires a much deeper mechanistic understanding of FORCP. Along these lines:i)Iidentification of FORCP interacting proteins might provide allow the authors to elucidate a molecular function for this peptide, which would significantly increase the impact of this work.ii) Alternatively, there is little insight here as to how FORCP functions in ER stress; the authors show that silencing FORCP leads to decreased induction of Bip and Chop, but this could be because silencing FORCP enhances cell death and these mRNAs are decreased. The manuscript ends with a bit of a disappointing note, and no hint as to the mechanism whereby FORCP works, in ER stress or in tumorigenesis or otherwise. Including clear data as to the mechanism whereby FORCP functions in ER stress or tumorigenesis would strengthen the impact of the work.

During the revision of our manuscript, we identified FORCP-interacting protein(s) by performing FORCP-FLAG pulldowns followed by mass spectrometry (N=4) from cell lysates. Of the 9 FORCP-interacting proteins we focused on BRI3BP, a transmembrane protein that is not well-characterized. Our rationale to focus on BRI3BP was based on some evidence in the literature that BRI3BP could have a function in ER stress (Christianson et al., 2011). We have gathered the following new data: (1) validation of the BRI3BP-FORCP interaction by co-IP, (2) colocalization of BRI3BP and FORCP in the ER using confocal microscopy, (3) during ER stress, knockdown of BRI3BP resulted in increased cell death whereas knockdown of FORCP resulted in decreased cell death, (4) concurrent knockdown on FORCP and BRI3BP significantly rescues cell viability during ER stress indicating a functional interaction between FORCP and BRI3BP. Overall, these results provide encouraging insights into the potential mechanism by which FORCP functions during ER stress.

However, determining the precise mechanism by which the FORCP-BRI3BP complex functions during ER stress will require many additional experiments.

As discussed with the editors and reviewers, we were not able to add this data due to space limitations.

2) The most important technical weakness of this paper is the use of only a single siRNA pool for all the FORCP phenotypic experiments. Particularly because very general phenotypes are being studied (proliferation, cell death), it is essential that the authors do a better job of establishing the specificity of these results. Ideally, CRISPR would be used to knockout FORCP to confirm these findings. Although it may be difficult to grow the well-differentiated CRC cell lines as single cell clones, CRISPR knockouts can be highly effective in lentivirally-infected pools, so this is not a sufficient reason to avoid genome-editing experiments. Alternatively, it might be acceptable to use additional siRNAs and perform rescue experiments, but use of CRISPR would be more convincing. The authors would need to repeat all the experiments in Figures 6-7 with this additional loss-of-function approach.

We thank the reviewers for this great point. As suggested, we used the CRISPR/Cas9 technology to knockout (KO) *FORCP* in the well-differentiated LS180 cells. We found that, similar to our FORCP siRNA experiments, as compared to the FORCP-WT cells, isogenic FORCP-KO cells proliferate faster under untreated conditions and undergo decreased cell death during ER stress. These data validate the specificity of our results using siRNA pool against *FORCP*. Using the *FORCP*-WT and isogenic *FORCP*-KO LS180 cells, we have repeated the majority of the experiments, as suggested by the reviewers. The new data are presented in Figure 4D-H and Figure 4—figure supplement 3A-D of the revised manuscript.

3) The in vivo tumor growth data are lacking a clear description and appear to have been performed in non-conventional manner. Injecting 1 million cells embedded in Matrigel in the flank and reporting one volume measurement after 3 weeks does not provide sufficient detail into the effects of the studied protein. These studies must be performed and prevented more conventionally. In flank tumor models, tumor growth can be extended up to 1,000-1,500 cc and measurements on a daily or every other day should be reported. It is possible that the tumors did not grow at all, or grew very little as the data is presented. More importantly, there are no correlative studies to test the main findings in vitro, e.g., staining for Ki-67 or EdU incorporation (proliferation) and cleaved caspase-3/TUNEL (apoptosis) in the tumors. The inhibition of FORCP before injection of the tumors could very well have affected angiogenesis and not proliferation or resistance to apoptosis.

Having generated the FORCP-KO cells, we were planning to do this in vivo experiment. However, we had to abruptly terminate this experiment due to the pandemic. Given that we would need at least 3 months to do this experiment and the uncertainty of the current situation, the editors and reviewers have kindly agreed to our request of not including these data.

4) Mechanistically, the authors show that in vitro, FOXA1 binds to the promoter and an intronic region of FORCP gene. However, it is unclear if FOXA1 is required for the ER stress-dependent induction of FORCP.

During the revision, we have addressed this question. Our data indicate that FOXA1 may not be necessary for inducing FORCP expression during ER stress (Figure 4—figure supplement 1C).

5) In terms of ER stress, only two acute and non-physiological stresses, DTT and Tunicamycin were used. Since the authors suggest a tumor-suppressive role of FORCP, they should try more relevant stresses such as hypoxia and amino acid or glucose deprivation.

We have addressed this concern by conducting new experiments as suggested. Our data indicate that in the context of other stresses such as hypoxia and inhibition of glucose metabolism, FORCP-KO cells show significantly better than FORCP-WT cells (Figure 4—figure supplement 3C and D). The difference in survival was modest but significant.

[Editors' note: further revisions were suggested prior to acceptance, as described below.]

Essential revisions:1) I do not think that the polysome fractionation data in Figure 2B is interpretable without the A260 absorbance data, which is necessary to determine which fractions contain polysomes. As the data are presented now, this is just a sucrose gradient fractionation, showing that FORCP RNA associates with fractions in the middle of the gradient. FORCP could be polysome associated, or it could be associated with some other EDTA-sensitive complex. Given the weakness of these data, and the authors' inability to perform the appropriate analysis due to the pandemic, I suggest that these data be removed. There are sufficient other data to show that FORCP encodes a protein without this experiment.

We thank the reviewers for this comment and have removed the polysome data from the paper.

2) The images in Figure 3C, which show endogenous FORCP protein, do not very convincingly show co-localization with PDI (there is clearly some co-localization, but the FORCP distribution appears to be broader). Quantification of co-localization in a larger number of cells should be provided.

We agree and have added the quantification of co-localization for FORCP and PDI in the revised Figure 3C.

3) The inclusion of additional, more physiological stressors in the experiments described in Figure 4 is a positive development. However, the use of CoCl_2_ in lieu of "true hypoxia" (i.e., elicited by use of an environmental hypoxia chamber), is problematic. CoCl_2_ "mimics hypoxia" only as far as induction of HIF-1a or HIF-2a is concerned, via inhibition of the activity of prolyl-hydroxylases and post-translational stabilization of those proteins. Induction of ER stress and the UPR by hypoxia depends on actual reduction in intracellular O_2_ levels leading to accumulation of unfolded proteins. Therefore, Figure 4—figure supplement 3C should be removed and only the low glucose data be shown. A more detailed analysis on the role of FORCP in cell survival under hypoxia should be re-examined in more detail later.

We agree and have removed this data.

4) The authors describe a proteomic study that identifies a protein associated with ER stress, BRI3BP. At the very least, presenting the IP-western indicating a functional interaction would provide some hint as to the mechanism whereby FORCP functions in ER stress; the authors would only have to present the proteomics and interaction and could save the remaining new data for a more comprehensive functional study.

We agree with the reviewers’ comment. The new data has been added in revised Figure 4, Figure 4—figure supplement 4 and in Supplementary file 4. Briefly, we have included the names of all proteins that were enriched in the FORCP-FLAG pulldowns (N=4) followed by mass spectrometry (Supplementary file 4). A shorter list after removing potential false positives using CRAPOME is provided in Figure 4G. As suggested, we have validated the interaction between FORCP and BRI3BP by IP-Western (Figure 4H). Finally, we show that like FORCP, BRI3BP has a role in regulating apoptosis in response to ER stress (Figure 4I). These results provide hint to the mechanism by which FORCP functions during ER stress.